# Process-Oriented Instrument and Taxonomy for Teaching Surgical Procedures in Medical Training: The Ultrasound-Guided Insertion of Central Venous Catheter [note 1]

**DOI:** 10.3390/ijerph17113849

**Published:** 2020-05-29

**Authors:** Victor Galvez, Rene de la Fuente, Cesar Meneses, Luis Leiva, Gonzalo Fagalde, Valeria Herskovic, Ricardo Fuentes, Jorge Munoz-Gama, Marcos Sepúlveda

**Affiliations:** 1Department of Computer Science, School of Engineering, Pontificia Universidad Católica de Chile, Santiago 7820436, Chile; vagalvez@uc.cl (V.G.); cnmeneses@uc.cl (C.M.); lileiva@uc.cl (L.L.); gfagalde@uc.cl (G.F.); vherskov@ing.puc.cl (V.H.); jmun@uc.cl (J.M.-G.); marcos@ing.puc.cl (M.S.); 2Department of Anesthesiology, School of Medicine, Pontificia Universidad Católica de Chile, Santiago 8331150, Chile; rfuente@med.puc.cl

**Keywords:** medical training, surgical procedures, Process Mining, medical instructor, procedural skills

## Abstract

Procedural training is relevant for physicians who perform surgical procedures. In the medical education field, instructors who teach surgical procedures need to understand how their students are learning to give them feedback and assess them objectively. The sequence of steps of surgical procedures is an aspect rarely considered in medical education, and state-of-the-art tools for giving feedback and assessing students do not focus on this perspective. Process Mining can help to include this perspective in this field since it has recently been used successfully in some applications. However, these previous developments are more centred on students than on instructors. This paper presents the use of Process Mining to fill this gap, generating a taxonomy of activities and a process-oriented instrument. We evaluated both tools with instructors who teach central venous catheter insertion. The results show that the instructors found both tools useful to provide objective feedback and objective assessment. We concluded that the instructors understood the information provided by the instrument since it provides helpful information to understand students’ performance regarding the sequence of steps followed.

## 1. Introduction

Medical education aims to prepare physicians with the latest scientific discoveries in the prevention and treatment of illnesses and diseases that people suffer [1]. Surgical procedures are an essential part of these treatments, and doctors need to be proficient in procedural skills to perform them successfully. Procedural skills (i.e., skills needed to perform surgical procedures) are one of the technical competencies considered when teaching [2], and they are relevant because they are associated with good clinical outcomes [3].

An instructor who teaches procedural skills needs to understand student performance regarding the sequence of steps of a surgical procedure. With this in mind, instructors can develop strategies to provide specific feedback and assess their students, which are competencies an instructor should have [4].

Currently, instructors teach procedural skills using several tools. Standard tools are checklists with the steps needed to complete a procedure [5], and Global Rating Scales (GRS) [6], which help to qualitatively assess indicators such as the flow of the procedure and economy of movements [5]. Tools mentioned above are useful, but state-of-the-art tools rarely take into account the sequence of steps: they are focused on each isolated step and do not consider the relative importance of each step or the incorrect execution of the sequence of steps.

Process Mining [7] is a new discipline that allows the analysis of processes using data stored and generated by information systems that support them. This discipline has been used successfully in a wide variety of healthcare specialities [8], among them medical education. Lira et al. [9] showed the use of Process Mining to give specific feedback, using data obtained from recorded executions of a procedure performed by students. In addition, de la Fuente et al. [10] used Process Mining to compare trainees and experts with the ideal sequence of steps. However, neither study emphasized information relevant to instructors, and they are difficult for them to interpret.

Surgical procedures are a progression of steps [11] that can be seen as a process [12]. Therefore, in this research, we used Process Mining to help instructors to understand students’ performance regarding the sequence of steps in surgical procedures. The information captured by an analysis of the sequence of steps through Process Mining adds information that is not possible to capture with a checklist, which is limited to a dichotomous assessment of the execution of specific steps of a procedure, without considering the order or the unnecessary repetition. Conversely, it explicitly explains a qualitative dimension included in the GRS, such as fluidity in the execution of the steps of a procedure.

The objectives of this research were: (1) to define a taxonomy of steps for surgical procedures; (2) to design an instrument for instructors with process-oriented information about the sequence of steps followed by their students; and (3) to evaluate with instructors the usefulness of both tools for their tasks as teachers. The approach presented has two steps: first, we developed the taxonomy, and then we designed the process-oriented instrument based on the taxonomy.

This article is an extension of our previous workshop article [13]. Here, we add: (1) information about current ways instructors use to teach the sequence of steps; (2) an evaluation of the taxonomy’s usefulness through open-ended questions; (3) an evaluation of the instrument’s interpretability by means of a questionnaire; (4) an evaluation of the instrument’s usability using the System Usability Scale (SUS) [14]; and (5) the opinion of instructors about the instrument after using it.

The structure of this paper is as follows. First, we describe the running case used to generate the taxonomy and the instrument. Second, we detail the taxonomy of activities and questions generated to discover undesired patterns. Third, we explain the development of the instrument and how it answers questions generated before. Fourth, we describe the current methods and tools used by instructors to teach the sequence of steps, evaluate the usefulness of the taxonomy, and evaluate interpretability and usability of the instrument. Finally, we present our conclusions and future work.

## 2. Running Case: The Ultrasound-Guided Internal Jugular Central Venous Catheter Placement

We used the Ultrasound-Guided Internal Jugular Central Venous Catheter (UGIJCVC) placement procedure as a running case to illustrate our approach. UGIJCVC consists of the installation of a tube in the central vein, to assist in the delivery of fluids or medications to a patient. This procedure has the following steps:Prepare implements, set up the ultrasound equipment, and position the patient.Identify the target vein with the ultrasound, and then puncture it with a trocar. A trocar is a needle with a hole to insert the guidewire.Verify blood return using a syringe. If it happens, the trocar was correctly installed. Then, remove the syringe.Pass a guidewire through the trocar. Once the guidewire is in the vein, remove the trocar.Widen the pathway and insert the catheter using the installed guidewire.Remove the guidewire and install the catheter.

Essential materials to do a process-oriented analysis of the UGIJCVC case were a process model (modeled using BPMN notation) and an event log with executions of the process [15].

To generate the process model, an initial model was generated using activities included in validated checklists. To avoid a biased mode, de la Fuente [16] developed a Delphi panel with experts on the procedure through an online survey. They included the activities in the model when the panel reached 80% consensus for each activity, thus obtaining the ideal execution of the procedure.

Data used correspond to executions performed by ten residents of a simulation-based training course at Pontificia Universidad Católica de Chile [17], where students received training on the UGIJCVC placement, and students enrolled in the course were given process-oriented feedback [9]. We obtained data using a web-based software called POMElog [18]. This software allowed us to generate event logs from videos recorded while students performed the procedure. Each event of the event log used contains a student as case identifier, a procedure step as activity, and date and time when the student performed the procedure step as the timestamp. An expert manually tagged each video with the activities shown in the process model (see Figure 1). Once all videos were tagged, we obtained the event log used in this article.

This study has the approval (ID: 16-194) of the Pontificia Universidad Católica de Chile ethics research committee.

## 3. Taxonomy of Activities

The first contribution of this paper is a taxonomy for a procedural skills training course. Instructors need to understand the performance of their students easily, but procedures commonly have many steps, and the sequence of steps followed by each student is different. Therefore, information about how students are learning the sequence of steps could be challenging to understand for instructors. With a taxonomy of activities, it is possible to label the steps with a specific category, produce more synthesized information, and analyze students’ performance more easily.

We generated a taxonomy using the semantic of the BPMN model provided by Munoz-Gama et al. [15]. This semantic is related to the domain where the model is applied, represents what surgeons think when they perform surgical procedures, and allows us to classify the activities (i.e., steps) of this model in the following four categories:**1.** 
**Preparation** activities.Steps previous to the beginning of the procedure. These steps correspond to the preparation of the patient and implements needed for the execution of the surgical procedure.**2.** 
**Identification** activities.Steps to recognize and locate a structure (e.g., vein and lung) that will be intervened during an action activity. The execution of these steps is always before an action activity.**3.** 
**Action** activities.Main steps of a surgical procedure. They represent steps that indicate progress along the stages of the procedure.**4.** 
**Control** activities.Steps to verify the correct execution of an action activity or check if the objective of the action activity was accomplished. Thus, they define if it is possible to continue with the next step or they should go back. They are always after an action activity.**5.** **Other** activities.Steps not performed in a simulation context or steps that make no sense categorizing in one of the four categories mentioned.

We classified activities depending on the task performed: activities that help to know what is needed before performing a procedure (preparation activities), locate the structure (identification activities), execute a main step of the procedure (action activity), and check if the step was done correctly (control activities).

This definition considers that each activity can belong to only one category. In case it is not clear what activity class a step belongs to, it is possible to split it into more steps, and then classify them in any of the proposed categories.

Figure 1 shows the BPMN model with the proposed taxonomy applied. The first 11 steps are preparation activities, including hand-washing and patient positioning. Then, identification activities such as ‘Doppler identification’ help to determine where to ‘Puncture’, which is an action activity. After, ‘Blood return’ is the control activity to verify if ‘Puncture’ was done correctly (i.e., to check if the trocar is inside the vein). Later, ‘Guidewire install’ and ‘Remove trocar’ are the next action activities to execute, to then be verified by ‘Check wire in long axis’, ‘Check wire in short axis’, and ‘Wire in good position’, all control activities. Finally, ‘Advance catheter’ and ‘Remove guidewire’ are the last action activities, which should be controlled by ‘Check flow and reflow’ and ‘Check catheter position’, both control activities.

## 4. Discovering Undesired Patterns

In this section, we present the design of an instrument (see Figure 2) to discover undesired patterns. This instrument helps to do a retrospective analysis of the course performance regarding sequence errors. In addition, the instrument provides information at the course level and at the specific student level, allowing the comparison between both levels, knowing the overall performance of the course and thus planning the next sessions, with either current or future students.

The instrument contains answers for questions designed using the taxonomy presented in Section 3. We remark that these questions did not allow us to analyze the successfulness of each isolated step, but helped us to discover undesired patterns related to the sequence of steps. Below, we present the four questions, each with its answer. We also generated the answers using the taxonomy, and then we put them together in the instrument shown in Figure 2.

Regarding Process Mining techniques, we used the ideas of algorithms based on Directly-Follows Graphs [19], which are commonly implemented in commercial tools (such as Disco and Celonis). We used this approach because it can be easily understood by non-expert users of Process Mining. Although this approach has some limitations such as a representative bias, it shows the behavior of a process in a simple way [19].

**Q1.** 

**What undesired sequence of action steps are students doing?**
Execution of surgical procedures consists of following a specific order of action steps correctly. Undesired patterns are going back on action steps, omitting action steps, or repeating unnecessarily an action step.**A1.** 

***Sequence of action activities executed.***
**A1** is the answer for **Q1**. For this answer, we used events in the event log that correspond to *action* activities. It allowed us to know where students went back, did repetitions of a step, or if they omitted any of the main steps of the procedure.For a student, the answer shows the number of times the student followed the path between two action activities, represented by the arrow. Figure 2 (see A1) shows the sequence executed by the student Peter, where is possible to view this student repeated once ‘Puncture’ and ‘Advance catheter’, went back once from ‘Guidewire install’ to ‘Puncture’, and also went back once from ‘Remove guidewire’ to ‘Puncture’. This student went from ‘Puncture’ to ‘Guidewire install’ three times, from ‘Guidewire install’ to ‘Remove trocar’ two times, from ‘Remove trocar’ to ‘Advance catheter’, and from ‘Advance catheter’ to ‘Remove guidewire’.For the course, the answer shows the number of students who followed the path between two action activities at least once. In Figure 2 (see A1), the view for the course shows that two students repeated ‘Puncture’ at least once, and one student repeated ‘Advance catheter’ at least once; two students went back from ‘Guidewire install’ to ‘Puncture’ at least once, and one student from ‘Remove guidewire’ to ‘Puncture’ at least once. The whole course (ten students) went from ‘Puncture’ to ‘Guidewire install’, from ‘Guidewire install’ to ‘Remove trocar’, from ‘Remove trocar’ to ‘Advance catheter’, and from ‘Advance catheter’ to ‘Remove guidewire’ at least once.**Q2.** 

**How many identification and control steps were executed?**
Some steps of surgical procedures involve the intervention of an organ or part of the body. Before intervening, it is essential to identify the organ, i.e., locate the structure that will be intervened. Other steps check the installation of an instrument or verify if an action had the expected result (e.g., positioning the catheter or other instrument). An undesired pattern is the excessive execution of identification and control steps, because this indicates a lack of fluidity and economy of movement [6].**A2.** 

***Amount of identification and/or control activities executed.***
**A2** is the answer for **Q2**. For this answer, we used events in the event log that correspond to *action, identification*, and *control* activities.It allowed us to know if students are making an excessive or insufficient amount of identification and/or control activities that are related to each action activity.For a student, the answer shows the number of events that are identification activities (left side) and/or control activities (right side) for each action activity. In addition, in parenthesis is the desired amount of events. If there is nothing on the right or the left side of the action activity, it means it is not necessary to perform the absent side. Figure 2 (see A2) shows the number of activities executed by student Peter. The student did two times identification activities before ‘Puncture’ and four times control activities, but the desired amount is once. Peter did twelve times control activities of ‘Guidewire install’ and ‘Remove trocar’, but the desired amount is two. Besides, Peter did once control activities after ‘Advance catheter’ and ‘Remove guidewire’, but the best approach is to perform two control activities.For the course, the answer shows the number of students who perform at least one identification (left side) and/or control (right side) activities for each action activity. If there is nothing on the right or the left side of the action activity, it means it was not necessary to perform the absent side. Figure 2 (see A2) shows that all students performed some activities before and after ‘Puncture’, seven students executed some activities after ‘Guidewire install’ and ‘Remove trocar’, and all students did some activities after ‘Advance catheter’ and ‘Remove guidewire’.**Q3.** 

**Were identifications and controls executed correctly?**
Besides the number of times executed, it is essential to know if students are identifying and controlling an action step correctly each time it is executed. An undesired pattern is not controlling or identifying each time an action step is executed.**A3.** 

***Identification and/or control each time an action activity is executed.***
**A3** is the answer for **Q3**. For this answer, we used events in the event log that correspond to *action, identification*, and *control* activities.It allowed us to know if students performed the identifications and/or controls required by the BPMN model each time students executed the action activity.For a student, the answer in Figure 2 (see A3) shows the number of times the action activity was executed. The number is accompanied by ‘✓’ when an identification and/or control was correctly done, or ‘✕’ if it was done incorrectly. A3 in Figure 2 shows identification (left side) and/or control (right side) for each action activity. If there is nothing on the right or the left side of the action activity, it means it was not necessary to perform the absent side. To determine the correctness, we created the following rules:We considered the identification as correct:·If the model defines only one identification activity before the action activity (see Figure 1), the event that corresponds to the identification activity should be directly followed by an event that corresponds to the action activity in the analysis.·If the model defines more than one identification activity (see Figure 1), the events should be executed in the way defined by the model.We considered the control as correct:·If the model defines only one control activity after the action activity (see Figure 1), an event that corresponds to the control activity should be just after an event that corresponds to the action activity in the analysis.·If the model defines more than one control activity (see Figure 1):-Events that correspond to control activities should be executed in the way defined by the model.-If only one control activity was executed after the action activity, an activity prior to the action activity in the analysis should be executed. This behavior shows that the control activity executed was enough for the student to realize the mistake done.Figure 2 (see A3) shows the results for student Peter. This student did ‘Puncture’ four times, but only performed the identification correctly the first time (1 of 4, 25%), and performed the control of ‘Puncture’ every time (4 of 4, 100%). Peter performed ‘Remove trocar’ two times and only performed the control correctly the first time (1 of 2, 50%). Besides, Peter performed ‘Remove guidewire’ two times, and did not do the control correctly both times.For the course, this answer shows the percentage of students who made the identification (left side) and/or control (right side) every time the action activity was executed. In Figure 2 (see A3), the view for the whole course shows that 90% of students made the identification of ‘Puncture’ every time it was executed, and 80% of them did the control correctly; 50% of students controlled ‘Remove trocar’ successfully every time it was performed; and 90% of students controlled ‘Remove guidewire’ successfully every time it was executed.**Q4.** 

**Are the students doing preparation steps during the execution of the procedure? Where?**
Before the execution of any surgical procedure, it is essential to perform previous steps to prepare the patient and the implements needed along with the procedure. An undesired pattern is to do preparation steps once the procedure begins because it indicates bad preparation by students in the performance of the procedure.**A4.** 

***Preparation activities during the execution.***
**A4** is the answer for **Q4**. For this answer, we used events in the event log that correspond to *action* and *preparation* activities.It allowed us to know if preparation activities were executed after an action activity. An ideal execution performs all the preparation activities at the beginning (as is stipulated in the BPMN model, see Figure 1). An undesired execution shows preparation activities executed between action activities, and it indicates that the student did not prepare the procedure or the patient correctly.For a student, the answer shows the number of events that correspond to the preparation activities executed after each action activity. Figure 2 (see A4) shows the results for student Peter and indicate that the student did preparation activities after ‘Guidewire install’ and ‘Advance catheter’ once.For the course, the answer shows the number of students who performed preparation activities at least once after each action activity. In Figure 2 (see A4), the view for the whole course shows four students did preparation activities after ‘Puncture’ at least once, three students after ‘Guidewire install’, and one student after ‘Advance catheter’.

## 5. Evaluating the Taxonomy and Instrument with Instructors

We conducted the evaluation through the following stages shown in Table 1. The focus of the first stage was to know the current ways instructors use to teach the sequence of steps, give feedback about the sequence, and assess whether students learned the sequence. Then, we evaluated the ease of understanding and usefulness of taxonomy for typical instructor tasks, and the interpretability and usability of the instrument as well as the opinion of instructors after using the instrument with the instructors.

For evaluating the taxonomy and the instrument, we asked three experts who commonly teach UGIJCVC placement. They teach in two institutions in anesthesiology and internal medicine specialities, have 5.7 years of experience on average as UGIJCVC instructors, and 12.3 years of experience performing the procedure on average. Evaluation results were analyzed using qualitative content analysis [20]. We asked instructors to answer questionnaires written in a paper, and open-ended questions were recorded. Then, we transcribed the audio and analyzed them, to then create categories coding quotes to interpret the answers. We included participant quotes in the paper, and we identified them anonymously with I1, I2, and I3.

### 5.1. Current Teaching of the Sequence of Steps

We asked instructors to describe how they teach the sequence of steps, give feedback to students, and how they assess the sequence of steps. Questions answered by instructors are in Table 2.

Answers to Question 1 of Table 2, regarding methods of teaching, varied between all the participants’ instructors. One instructor said “I give myself as an example of how the process is carried out” (I1). Two instructors said they teach the sequence partitioning the procedure in stages. In addition, one instructor encourages students to verbalize what they are doing for two reasons: one is “the assistant (...) needs to know what is the doctor doing” (I3), and second “we want the student not only to learn the technique but also to lead the procedure” (I3). Regarding tools, two instructors used a checklist of the procedure, based on the literature. Instructors used it to indicate the steps using the order predefined by the checklist. One instructor said she teaches “mentioning the steps a little with the checklist in mind but not with the paper in hand” (I2). Another instructor sends students videos of how to perform the procedure and documents with anatomic information weekly prior to training sessions.

Answers to Question 2 of Table 2, regarding giving feedback methods, noted that instructors do it without a specific structure or pattern. It is a problem for feedback effectiveness because the reliability and credibility of the feedback is an issue [21]. One instructor said “I feel that we are always weak in the feedback” (I1) because “we don’t have an objective way to correct mistakes” (I1). Regarding tools, two instructors said they give feedback using a global scale (which is a qualitative assessment and is useful for any procedure), a checklist or by advising students based on their experience performing the procedure. One instructor mentioned “a checklist is quite extensive but it is super meticulous for detail” (I3), and another instructor mentioned that she prefers global scales instead of a checklist because “checklists did not discriminate between experts and novices” (I2). The instructor who considers giving feedback challenging did not mention tools such as checklists or global scales, but said “we try to correct at the time the mistake is made” (I1).

Answers to Question 3 of Table 2, regarding methods of assessment, showed issues mentioned by instructors: “I think we do it in a super qualitative way” (I2), “there is no objective pattern” (I1), “We do not have full standardization between teachers” (I2), and “the checklist is long, it is very extensive, which makes it a bit difficult when you evaluate” (I3). Regarding tools, two instructors use the checklist and the global scale, but without putting the focus on the sequence of steps. One instructor assesses subjectively if the sequence executed was correct. Concerning the checklist, one instructor said “(the students were) very clear in saying they preferred a very detailed checklist step-by-step, because it was useful for them later in the formative part” (I3), and another mentioned that “today we do not have a tool or an instrument to evaluate that” (I2).

### 5.2. Taxonomy

We evaluated the taxonomy by asking instructors about dimensions shown on Table 3 using a five-point visual analogue scale, where one means ‘totally disagree’ and five ‘totally agree’. In addition, we asked instructors their opinion about the usefulness of the taxonomy for giving feedback and assessment tasks.

The results show that all the instructors agreed with Sentences 1 and 2 (see Table 3). In addition, two instructors totally agreed and one agreed with Sentence 3 (see Table 3). All the instructors think taxonomy can help them give feedback and assess procedural skills.

Comments of instructors regarding the taxonomy are “I try to think about how to improve or give it another classification and I can’t” (I2), “I think it is really good, the structure helps to carry out a more objective assessment” (I2), “I think that, this process model will be really useful in self-taught training in the future” (I2) and with this, one can say “hey, look you failed on this” or “you did well on this” and “the steps to follow are these” (I3).

### 5.3. Instrument Interpretability

After explaining the instrument to instructors, we asked them to answer an interpretability test, and thus we evaluated their understanding of the information given by the instrument. It consists of asking instructors questions about each answer given by the instrument. The instrument (see Figure 2) was generated using a real student and the course in [15]. Interpretability test questions are in Table 4.

The rsults show that all instructors answered the majority of questions on Table 4 correctly (88.1% correct answers on average by each instructor, SD = 3.37%), confirming the success of this Interactive Pattern Recognition case [22]. All instructors answered all the questions related to the number of identification/control activities executed and procedure preparation correctly (Answers A2 and A4, respectively). However, instructors answered some questions regarding repetitions or reworks of action activities incorrectly (two instructors answered Question 1.1 incorrectly and one instructor answered Question 1.3 incorrectly), as well as questions related to the correct execution of identification and control (one instructor answered Question 3.2 incorrectly and another instructor answered Question 3.3 incorrectly).

### 5.4. Instrument Usability

Usability was defined by Brooke [14] as the appropriateness of an artefact for a specific purpose. Regarding Point 5 of Table 1, we used the System Usability Scale (SUS) to determine usability of our instrument [14], a widely accepted questionnaire to evaluate it. SUS questions are in Table 5. Instructors answered each question using a five-point visual analogue scale, where one means ‘strongly disagree’ and five ‘strongly agree’.

The mean score was 89.2 (SD = 9.2). According to Bangor et al. [23], it means our instrument has an acceptable level of usability. Analyzing the questions, it is possible to see that all instructors agreed or strongly agreed about the frequency with which they would use the instrument, ease of use, integration of the components, fastness of learning to use the instrument, and confidence using the instrument (Questions 1, 3, 5, 7 and 9 in Table 5). All instructors disagreed or strongly disagreed regarding the complexity of the instrument, inconsistency in the instrument and difficulties using the instrument (Questions 2, 6, and 8 in Table 5). One instructor agreed with the need for expert support to use the instrument while two instructors disagreed or strongly disagreed with this sentence (Question 4 on Table 5). The question related to the need for learning things before using the instrument had the same trend (Question 10 on Table 5).

### 5.5. Instructors Opinion after Using the Instrument

To describe the opinion of instructors about using the instrument, we asked them open-ended questions after they used it to answer the interpretability test (see questions in Table 6). In addition, we asked instructors about the usefulness of the instrument giving feedback and assessment tasks.

Regarding Question 1 of Table 6, instructors said the instrument helps to do a more objective assessment, because even “one skip steps (performing the procedure), and oneself also skips them when evaluating” (I1), which is an issue supported by the literature [24]. In addition, one instructor said “this can help us know mentally if the student knows the next step, both the instructor and the student” (I2) and another instructor mentioned the “different items of the instrument served to understand that sequence” (I3).

Answering Question 2 of Table 6, instructors highlighted the characteristics of the instrument: one instructor mentioned “the comparison with the course is always more attractive than the individual as a single entity” (I3). In addition, they mentioned the possibility of tasks they can do with the instrument in comparison with others: one instructor said “one can compare with previous years, with other groups and with other groups from other institutions” (I1), “it is often challenging to train the person to use the global scale, I think this will be easier to train in because it is very logical” (I2), and “I would imagine that showing this to a student would make it easier for him/her to understand why it is important and why I am giving him/her this feedback or mark” (I3). Additionally, the instrument allowed instructors to reflect on their performance as a teacher: one instructor said “(the instrument) lets me know as an instructor if there is a step that I am not explaining well or there is something that needs more reinforcement” (I3), “I think it is exciting to know where I am strong and where I need to put more emphasis on preparation on the course level as well as on the individual level” (I3).

Answering Question 3 of Table 6, two instructors said the first component was the hardest to understand, but, once we explained it to them, the items were easy to understand. Regarding Question 4 of Table 6, a suggestion was to create material to facilitate the instrument interpretability (e.g., a video with an explanation of how to interpret it).

Additionally, all instructors thought the taxonomy could help them give feedback and assess procedural skills.

## 6. Discussion

In this paper, we present a taxonomy of steps for surgical procedures and an instrument for instructors, both focused on teaching the sequence of steps required to perform a surgical procedure. Both tools help to obtain process-oriented information that could be useful for surgical instructors during training. The taxonomy developed can be used by instructors in everyday tasks, such as giving feedback and assessing students. Instructors can do the same tasks using the instrument, which shows information related to mistakes in the sequence of steps through the components it provides.

Instructors agreed that the taxonomy could help to give a structure to training, and they found it useful for giving feedback and assessing students’ performance. Regarding the instrument, instructors understood the information provided and considered the usability of the instrument acceptable. Similar to taxonomy, instructors thought the instrument helped them establish if students knew the correct sequence of steps, and it could be useful for everyday tasks they do as surgical teachers. Hence, we conclude that the information provided by the instrument (generated using the taxonomy) could be useful to understand students’ performance as regards the sequence of steps. Therefore, the taxonomy and the instrument are resources to include in the tools instructors have to teach surgical procedures.

The taxonomy and the instrument can help instructors in building or enhancing their mental model. For instructors, it is challenging to share their mental model [25,26], and that means that they omit close to 70% of the information needed by students during their learning process [24]. The taxonomy and the instrument help to address these problems: with the taxonomy, it is possible to share information about the steps of the procedure and the correct sequence of them; with the instrument, the information utilized to teach the correct sequence is standardized, avoiding the omission of information. In addition, both tools help to give a structure as to how to teach the procedure, providing standardization of the contents and an objective way for assessment and giving feedback.

A limitation of this research is the number of instructors who participated in the evaluation. However, we asked instructors from different specialities and institutions; thus, we believe the number of instructors provides sufficient evidence to accomplish the objective of this research. Another limitation is the possibility of difficulties to handle the instrument if the number of steps of a procedure is more extensive than UGIJCVC placement. In that case, figures of answers generated (see Figure 2) could change a bit to provide the same information as could be needed. Furthermore, the instrument does not contain subjective information as provided by GRS, because the instrument shows objective information. Such a need could be addressed using the instrument as a complement to checklists and GRS, allowing the instructor to capture subjective and objective information.

Future work to enhance interpretability of the instrument are improvements regarding repetitions or reworks of action activities (A1 of Figure 2), the excessive amount of identifications and control activities performed by the course in (A2 of Figure 2), and the correctness of identification and control each time an action activity is performed (A3 of Figure 2). We detected these improvement needs based on results obtained by instructors in the interpretability test. Another improvement needed is the classification between minor and major sequence errors, creating a good display of this information. In addition, further research is needed to demonstrate if the taxonomy and the instrument improve the learning of a surgical procedure (here, we did an evaluation using expert opinion [27]), to demonstrate if the instrument helps to improve feedback and assessment of students, and to make the generation of the instrument by potential users available and customizable.

## 7. Conclusions

We present a taxonomy of activities for surgical procedures and an instrument for instructors showing undesired sequence patterns, generated using Process Mining. After evaluation of both tools with experts, we found them as easy to understand, interpretable by instructors, and with an acceptable level of usability. Further studies should be done once instructors gain experience using the tool.

## Figures and Tables

**Figure 1 ijerph-17-03849-f001:**
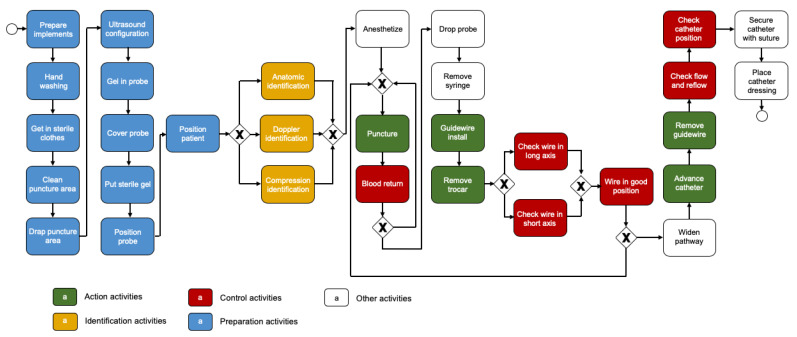
BPMN model of the UGIJCVC placement, enriched with the taxonomy proposed.

**Figure 2 ijerph-17-03849-f002:**
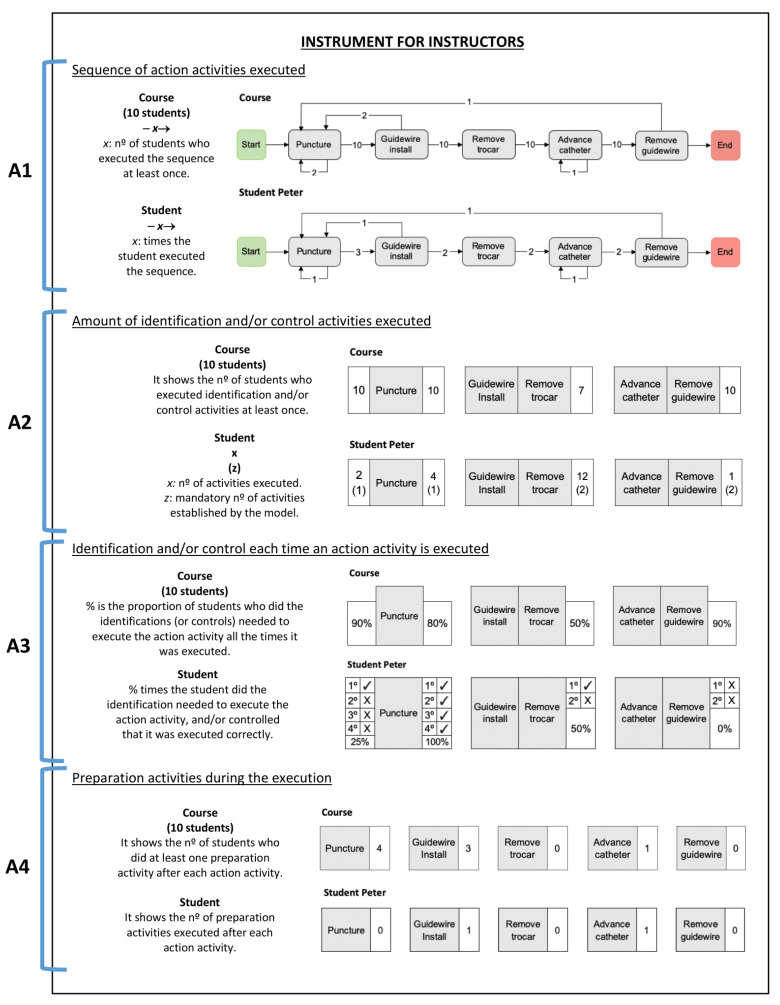
Process-oriented instrument for instructors of surgical procedures.

**Table 1 ijerph-17-03849-t001:** Evaluation stages.

Stage	Task
1	Current teaching of the sequence of steps.
2	Taxonomy explanation to instructors and its usefulness.
3	Instrument explanation to instructors.
4	Instrument interpretability by instructors.
5	Usability analysis of the instrument.
6	Instructors’ opinion after using the instrument.

**Table 2 ijerph-17-03849-t002:** Current teaching of the sequence of steps.

No.	Question
1	How do you teach the sequence of steps in the procedure currently? Do you use any tools or instruments?
2	How do you give feedback to students on their flow during the procedure currently? (For example, about what the next step to be performed is, if any are skipped, if it stops, etc.) Do you use any tools or instruments?
3	How do you assess students’ flow and the sequence of steps during the procedure currently? (For example, if the correct sequence of steps is executed, if any are skipped, if it stops, etc.) Do you use any tools or instruments?

**Table 3 ijerph-17-03849-t003:** Sentences to rate the taxonomy.

No.	Sentence
1	The taxonomy is easy to understand.
2	The taxonomy facilitates the development of a mental model of the procedure.
3	The taxonomy is applicable to other medical procedures (other than UGIJCVC).

**Table 4 ijerph-17-03849-t004:** Interpretability test questions.

Answer	No.	Question
A1	1.1	What activities did the course repeat?
	1.2	How many students did ‘Puncture’ after ‘Remove guidewire’?
	1.3	What activities did Peter repeat?
	1.4	How many times did Peter do ‘Puncture’ after ‘Remove guidewire’?
A2	2.1	How many students did the identification previous to ‘Puncture’?
	2.2	How many students did the control of ‘Guidewire install’ and ‘Remove trocar’?
	2.3	Did Peter do the ideal number of identification activities prior to ‘Puncture’? If you have a negative response, did he do more or less than the ideal number?
	2.4	Did Peter do the ideal number of control do activities after ‘Advance catheter’ and ‘Remove guidewire’? If you have a negative response, did he do more or less than the ideal number?
A3	3.1	What percentage of the course did the necessary identification each time they performed ‘Puncture’?
	3.2	What percentage of the course did the necessary control each time they performed ‘Advance catheter’ and ‘Remove guidewire’?
	3.3	Peter performed ‘Puncture’ 4 times. In which of them did he do the identification? In which of them did he do the control?
A4	4.1	How many students did preparation activities right after ‘Puncture’?
	4.2	How many students did preparation activities right after ‘Remove Guidewire’?
	4.3	How many activities did the student Ana do right after ‘Withdraw Trocar’?

**Table 5 ijerph-17-03849-t005:** System Usability Scale (SUS) to evaluate the usability of the instrument.

No.	Question
1	I think that I would like to use this instrument frequently.
2	I found the instrument unnecessarily complex.
3	I thought the instrument was easy to use.
4	I think that I would need the support of an expert to use this instrument.
5	I found the various components of this instrument were well integrated.
6	I thought there was too much inconsistency in this instrument.
7	I would imagine that most people would learn to use this instrument very quickly.
8	I found the instrument very difficult to use.
9	I felt very confident using the instrument.
10	I needed to learn many things before I used this instrument.

**Table 6 ijerph-17-03849-t006:** Questions to get instructors opinion about using the instrument.

No.	Question
1	Do you think the instrument helps you know if the students know the correct sequence of steps in the procedure? Explain briefly.
2	What is your opinion of this instrument, compared to the tools/instruments that you commonly use? Explain briefly.
3	What was the most difficult thing about using the instrument?
4	What would you improve about the instrument?

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
