# Peer review of "Process-Oriented Instrument and Taxonomy for Teaching Surgical Procedures in Medical Training: The Ultrasound-Guided Insertion of Central Venous Catheter†"

_ijerph, 2020, doi:10.3390/ijerph17113849_

Round 1

Reviewer 1 Report

Line 3- eliminate "in order."

Line 65- and then punctured it with the trochar.

Lines 64-71- List the procedures as actions instead of passive actions.

Line 126- The first 11 steps are preparation activities, including hand-washing and patient positioning.

Figure 1- I was surprised to read that securing the catheter was not addressed. Is the assumption that nursing personnel complete this?

Lines 210-213- Needs to be rewritten.

Lines 282-284- You asked instructors to provide quantitative data for a qualitative study? Perhaps this just needs to be reworded.

The authors should be commended on their focus, that being the instructors. Some practical questions that should have been asked include:

  1. How did the instrument assist the instructors with insuring their own proficiency with the skill. Since many instructors are not full-time faculty, the instrument may assist with their own knowledge of the step-by-step process.
  2. While the instrument appeared to identify the procedures, it could provide more practical information by identifying major v. minor errors. When teaching processes, lockstep conformity to an agreed upon process is often needed. There are times when identifying major v. minor errors will better determine a student's level of mastery. The instrument may help provide the instructor a guide to confirming these processes.

The reporting of the taxonomy of activities was confusing. For instance, when a student chooses additional punctures, it must be assumed that the attempt was "missed." This should start the procedure and the instructor's use of the instrument from the beginning. Perhaps the order of the taxonomy should be changes to meet the process?

Author Response

Line 3- eliminate "in order."

  • This line was modified, eliminating in final text “in order”:

Line 65- and then punctured it with the trochar.

  • This line was modified. The change was included on line 70 of the new version of the article, writing “and then puncture it with a trocar”.

Lines 64-71- List the procedures as actions instead of passive actions.

  • In the new version, we wrote the procedure list as actions instead of passive actions. We included the change between lines 69 - 76:

“1. Prepare implements, set up the ultrasound equipment and position the patient.

  1. Identify the target vein with the ultrasound, and then puncture it with a trocar. A trocar is a needle with a hole to insert the guidewire.
  2. Verify blood return using a syringe. If it happens, the trocar was correctly installed. Then, remove the syringe.
  3. Pass a guidewire through the trocar. Once the guidewire is in the vein, remove the trocar.
  4. Widen the pathway and insert the catheter using the installed guidewire.
  5. Remove the guidewire and install the catheter.”

Line 126- The first 11 steps are preparation activities, including hand-washing and patient positioning.

  • This line was modified. We included the change in line 134 of the new version of the article, writing “including hand-washing and patient positioning”.

Figure 1- I was surprised to read that securing the catheter was not addressed. Is the assumption that nursing personnel complete this?

  • The activities “Secure catheter with suture” and “Place catheter dressing” were considered in the initial model obtained through the Delphi panel. They were not included in the analysis because they were not part of the phantom training simulation sequence, a measure that is used to extend the useful life of this resource. We agree with the reviewer that it can cause confusion for the reader, so we decided to include them in Figure 1 in the category of "Other" activities, to improve understanding and avoid confusion in future readers.

Lines 210-213- Needs to be rewritten.

  • These lines were rewritten to improve readers' understanding. We included the change in lines 206 - 208:

“Figure 2 (see A2) shows all students performed some activities before and after `Puncture' (identification and control, respectively), seven students executed some activities after `Guidewire install' and `Remove trocar', and all students did some activities after `Advance catheter' and `Remove guidewire'.”

Lines 282-284- You asked instructors to provide quantitative data for a qualitative study? Perhaps this just needs to be reworded.

  • This line was modified. We included the change in lines 291 - 292 of the new version of the article, writing:

“We asked instructors to answer questionnaires writing in a paper, and open-ended questions were audio recorded.”

The authors should be commended on their focus, that being the instructors. Some practical questions that should have been asked include:

  1. How did the instrument assist the instructors with insuring their own proficiency with the skill. Since many instructors are not full-time faculty, the instrument may assist with their own knowledge of the step-by-step process.

  • We appreciate the reviewer's interesting proposal. We believe that while it is not easy to see how the instrument helps the instructor in his own procedural skills, our instrument is a contribution to the development of a mental model of procedure execution. Having a representation that describes successful execution step by step has been proposed as the initial step to any training of this type (Grantcharov, T.P et al; Clark, R. E., et col). For this reason, we have decided to include a paragraph referring to this topic in the discussion (lines 414 - 420) in the new version of the article:

The taxonomy and the instrument can help instructors in building or enhancing their mental model. For instructors is challenging to share their mental model [26,27], and that produce they omit close to 70% of the information needed by students during their learning process [25]. The taxonomy and the instrument help to address these problems: with the taxonomy, it is possible to share information about the steps of the procedure and the correct sequence of them; with the instrument, the information utilized to teach the correct sequence is standardized, avoiding the omission of information. Also, both tools help to give a structure as to how to teach the procedure, providing standardization of the contents and an objective way for assessment and giving feedback.”

Grantcharov, T. P., & Reznick, R. K. (2008). Teaching procedural skills. Bmj, 336(7653), 1129–1131. http://doi.org/10.1136/bmj.39517.686956.47

Clark, R. E., Pugh, C. M., Yates, K. A., Inaba, K., Green, D. J., & Sullivan, M. E. (2012). The Use of Cognitive Task Analysis to Improve Instructional Descriptions of Procedures. Journal of Surgical Research, 173(1), e37–e42. http://doi.org/10.1016/j.jss.2011.09.003

  1. While the instrument appeared to identify the procedures, it could provide more practical information by identifying major v. minor errors. When teaching processes, lockstep conformity to an agreed upon process is often needed.

There are times when identifying major v. minor errors will better determine a student's level of mastery. The instrument may help provide the instructor a guide to confirming these processes.

  • The main objective of this research was to use Process Mining to help instructors to understand students’ performance regarding the sequence of steps in surgical procedures, and thus generate useful information for instructors. In a previous publication (de la Fuente, R. et al), we tried to explore the information that can be obtained considering errors and adjusting to an ideal pattern of execution. For this reason, we believe that the information on errors related to violations of the ideal sequence and unnecessary repetitions are complementary in order to have a global vision of the student, and that they can help to meet other objectives such as measuring the mastery level. In particular, generating an instrument that differentiates between major and minor errors is part of future work, so we included the following in lines 433 - 434:

“Another improvement needed is to classify between minor and major sequence errors, creating a good display of this information”

de la Fuente, R., Fuentes, R., Munoz-Gama, J., Riquelme, A., Altermatt, F. R., Pedemonte, J., et al. (2020). Control-flow analysis of procedural skills competencies in medical training through process mining. Postgraduate Medical Journal, 96(1135), 250–256. http://doi.org/10.1136/postgradmedj-2019-136802

The reporting of the taxonomy of activities was confusing. For instance, when a student chooses additional punctures, it must be assumed that the attempt was "missed." This should start the procedure and the instructor's use of the instrument from the beginning. Perhaps the order of the taxonomy should be changes to meet the process?

  • We reordered points listed in lines 106 - 125 using the implicit order proposed by the taxonomy, and rewritten less wordy to avoid the confusion of the readers. Finally, the paragraph was left as follows in the final text:

  1. Preparation activities.

Steps previous to the beginning of the procedure. These steps correspond to the preparation of the patient and implements needed for the execution of the surgical procedure.

  1. Identification activities.

Steps to recognise and locate a structure (e.g. vein, lung) that will be intervened during an action activity. The execution of these steps is always before an action activity.

  1. Action activities.

Main steps of a surgical procedure. They represent steps that indicate progress along the stages of the procedure.

  1. Control activities.

Steps to verify the correct execution of an action activity or check if the objective of the action activity was accomplished. Thus, they define if it is possible to continue with the next step or should go steps back. They are always after an action activity.

  1. Other activities.

Steps not performed in a simulation context or steps that make no sense categorising in one of the four categories mentioned.

  • The proposed instrument is designed to be used retrospectively with information from the whole group that was trained, contrasting a specific execution against information from the whole group. For this reason, it is not available to the instructor from the beginning of the training.

Reviewer 2 Report

This study describes a method for instructors to have real information about how their students are learning a surgical technique. In this case, Ultrasound-Guided Insertion of Central Venous Catheter.
The method is adequately described, the controls presented meet the objective. The instructors have a positive evaluation of the tool. However, this number is limited.
No data are provided on its effectiveness against other techniques. This aspect should be included as a limitation (although it is obvious that they do not).No student opinions are collected. Neither, there is a measure of whether this procedure produces better learning.The Discussion section is absent. Authors does not compare this approach with other alternatives, presenting, theoretically, possible advantages or weaknesses. Lessons learned or how to implement the experience in other faculties/schools are not detailed.
The costs of the system are not described. This information would help to determine its viability elsewhere.
I have not been able to understand whether the system can anticipate the most common errors by redesigning tasks or introducing new instructions. Although I suppose this is the purpose.

It is not reported whether this design was favorably reported by a teaching or ethics research committee.

Verbal tenses must be unified, present and past tense are used interchangeably. The conventional rules for this type of study are to resort to the past.

Author Response

This study describes a method for instructors to have real information about how their students are learning a surgical technique. In this case, Ultrasound-Guided Insertion of Central Venous Catheter.

The method is adequately described, the controls presented meet the objective.

The instructors have a positive evaluation of the tool. However, this number is limited.

  • We agree with the reviewer that the number of instructors surveyed is limited. Aware of this limitation, we tried to increase the variability of respondents to decrease biases, ensuring at least more than one center and more than one medical specialty. In the discussion section, the number of respondents is addressed as a limitation of our work, on lines 421 - 423:

“A limitation of this research is the number of instructors who participated in the evaluation. However, we asked instructors from different specialities and institutions, and because of that, we believe the number of instructors provides sufficient evidence to accomplish the objective of this research.”

No data are provided on its effectiveness against other techniques. This aspect should be included as a limitation (although it is obvious that they do not). No student opinions are collected. Neither, there is a measure of whether this procedure produces better learning.

  • The objective of our research was to generate an instrument that would be useful for instructors from the perspective of the sequence of execution of steps of a procedure. In order to achieve that, we evaluated delivering information that was relevant and understandable for them, in addition to evaluating its usability through a standardized scale. Determining the effectiveness in learning and the assessment of the students requires a methodological confrontation that is beyond the objective of this work. However, a first approach to this objective was developed in a previous work, where we evaluated the satisfaction of students who received feedback based on the Process Oriented approach. This work was mentioned in lines 35 - 37:

“[9] shows the use of Process Mining to give tailored feedback to students in a pre-post scheme, using data obtained from recorded executions of procedures performed by students.”

  • We appreciate the comment and hope to elucidate the impact of the instrument on achieving learning goals and taxonomy in future research. We included this as a section of the discussion where we tackle future work, in lines 434 - 437:

“Also, further research is needed to demonstrate if the taxonomy and the instrument improve the learning of a surgical procedure (here we did an evaluation using expert opinion [28]), to demonstrate if the instrument helps to improve feedback and assessment of students, and to make the generation of the instrument by users available and customizable.”

The Discussion section is absent. Authors does not compare this approach with other alternatives, presenting, theoretically, possible advantages or weaknesses. Lessons learned or how to implement the experience in other faculties/schools are not detailed.

  • We appreciate the comment about the absence of a Discussion section, which can be confusing for the reader. Thus, we have decided to include this section and shorten the Conclusion section to a paragraph that concisely summarizes the result of our research.

  • The discussion includes a synthesis of the results obtained, how the taxonomy and the instrument would help the instructors, limitations and future work. To the best of our knowledge, we do not know of other tools that analyze the order of execution of a medical procedure or standardize the content of feedback for procedural training. However, the article includes quotes in which instructors report advantages compared to other tools, which is in lines 381 - 386. Additionally, we added a limitation of our instrument when contrasting it with the assessment tools currently in use in part of the discussion (lines 426 - 428). We included this as a discussion in lines 400 - 437:

“In this paper, we present a taxonomy of steps for surgical procedures and an instrument for instructors, both focused on teaching the sequence of steps required to perform a surgical procedure. Both tools help to obtain process-oriented information that could be useful for surgical instructors during training. The taxonomy developed can be used by instructors in everyday tasks, like giving feedback and assessing students. Instructors can do the same tasks using the instrument, which shows information related to mistakes in the sequence of steps through the components it provides.

Instructors agreed that the taxonomy could help to give a structure to training, and they found it useful for giving feedback and assessing students' performance. Regarding the instrument, instructors understood the information provided and considered the usability of the instrument acceptable. Similarly to taxonomy, instructors thought the instrument helped them establish if students knew the correct sequence of steps, and it could be useful for everyday tasks they do as surgical teachers. Hence, we conclude that the information provided by the instrument (generated using the taxonomy) could be useful to understand students' performance as regards the sequence of steps. Therefore, the taxonomy and the instrument are resources to include in the tools instructors have to teach surgical procedures.

The taxonomy and the instrument can help instructors in building or enhancing their mental model. For instructors, it is challenging to share their mental model [26, 27], and that means that they omit close to 70% of the information needed by students during their learning process [25]. The taxonomy and the instrument help to address these problems: with the taxonomy, it is possible to share information about the steps of the procedure and the correct sequence of them; with the instrument, the information utilized to teach the correct sequence is standardized, avoiding the omission of information. Also, both tools help to give a structure as to how to teach the procedure, providing standardization of the contents and an objective way for assessment and giving feedback.

A limitation of this research is the number of instructors who participated in the evaluation. However, we asked instructors from different specialities and institutions, and because of that, we believe the number of instructors provides sufficient evidence to accomplish the objective of this research. Another limitation is the possibility of difficulties to handle the instrument if the number of steps of a procedure is more extensive than CVC installation. In that case, figures of answers generated (see Figure 2) could change a bit to provide the same information as could be needed. Furthermore, the instrument does not contain subjective information as provided by GRS, because the instrument shows objective information. Such a need could be addressed using the instrument as a complement to checklists and GRS, allowing the instructor to capture subjective and objective information.

Future work to enhance interpretability of the instrument are improvements regarding repetitions or reworks of action activities (A1 of Figure 2), the excessive amount of identifications and control activities performed by the course in (A2 of Figure 2), and the correctness of identification and control each time an action activity is performed (A3 of Figure 2). We detected these improvements needs based on results obtained by instructors in the interpretability test. Another improvement needed is the classification between minor and major sequence errors, creating a good display of this information. Also, further research is needed to demonstrate if the taxonomy and the instrument improve the learning of a surgical procedure (here we did an evaluation using expert opinion [28], to demonstrate if the instrument helps to improve feedback and assessment of students, and to make the generation of the instrument by potential users available and customizable.”

  • We included a small paragraph with conclusions after completing the study in lines 440 - 443:

“We presented a taxonomy of activities for surgical procedures and an instrument for instructors showing undesired sequence patterns, generated using Process Mining. After evaluation of both tools with experts, we found them as easy to understand, interpretable by instructors and with an acceptable level of usability.”

The costs of the system are not described. This information would help to determine its viability elsewhere.

  • The evaluation of costs is part of future work, since it is a variable to consider when making the instrument available for anyone to use. We included this in future work (lines 436 - 437):

“... to make the generation of the instrument by potential users available and customizable.”

I have not been able to understand whether the system can anticipate the most common errors by redesigning tasks or introducing new instructions. Although I suppose this is the purpose.

  • The purpose of the system is to understand undesired patterns in the sequence of execution of steps, but a single application does not offer enough information to globally identify which are the most frequent errors. After using it repeatedly or with a larger number of students, one could identify persistent error patterns. The tasks suggested in this section require a greater amount of information, so after occupying the instrument on a larger sample, it may be possible to perform them reliably.

It is not reported whether this design was favorably reported by a teaching or ethics research committee.

  • We regret not having made explicit the approval of the ethics committee, which we now have done in lines 92 - 93. We also enclose the approval of the ethics committee.

“2.2. Ethical approval authority

This study has the approval (ID: 16-194) from the Pontificia Universidad Católica de Chile ethics research committee.”

Verbal tenses must be unified, present and past tense are used interchangeably. The conventional rules for this type of study are to resort to the past.

  • The document has been extensively revised by a native speaker translator, with an emphasis on better grammar and consistency in verb tenses used. These changes have been incorporated throughout the article.

Reviewer 3 Report

This paper focuses on the use of process mining to support medical training. It presents both a taxonomy of activities and a process-oriented instrument to provide feedback to students. The artefacts are evaluated by instructors who teach one particular procedure: the insertion of a central venous catheter.

After a thorough assessment of the paper, I can conclude that this paper constitutes a valuable contribution to the special issue. It covers a topic which translates process mining techniques to instruments which can be used in practice (in this case educational practice). There is a need for research in that direction. Even though the evaluation is based on the input of three instructors, I feel that this provides sufficient evidence that the proposed artefacts are valuable and, hence, should be developed further in the future. The only thing I miss in the current paper is a short subsection outlining which process mining techniques are used and how/to which extent the proposed instrument is actually available for use. Some other comments are listed below on a section-by-section basis, but these are not of a fundamental nature. I am confident that the authors can fix them within a reasonable timespan.

DETAILED REMARKS ON A SECTION-BY-SECTION BASIS

** Introduction **

- The authors might want to make it more explicit how process mining extends tools such as checklists or GRS in the fourth paragraph of the introduction. Adding an example of such an existing tool later in the paper could also help to understand the added-value of the proposed approach.

** Running case **

- Line 75-77: Readability would be enhanced if the sentence regarding event logs would be part of the same paragraph as the explanation how these event logs were created.

- Line 81-82: How does the web-based software work? Is it based on manual labelling?

** Taxonomy of activities **

- Line 87-92: The goal and benefits of the taxonomy of activities should be clarified.

- Line 97-117: An implicit order between activities is highlighted: preparation > identification > action > control. Wouldn’t it make sense to order them in the numbered list this way?

- This section would benefit from a bit of background information regarding how the flowchart in Figure 1 was created and validated.

** Questions to discover undesired patterns **

- Line 144-145: Another undesired pattern would involve executing action activities in a sequential ordering relation in the wrong order. This is currently missing.

- Line 152: It might be good to highlight why the excessive execution of identification/control steps is undesired.

- The authors could consider to integrate sections 4 and 5, in which the current section 4 would become a subsection. Currently, it is not entirely clear what the purpose of section 4 as an autonomous section is.

** Discovering undesired patterns **

- Line 171: Maybe the authors should elaborate a bit more about what the instructor can learn at the two levels.

- Line 206-207: At the course level, the measure indicates the number of students who performed at least one identification/control activity. Is this entirely consistent with Q2, where it is highlighted that an undesired pattern is the excessive execution of identification/control activities?

- It would be interesting to know to what extent the proposed instrument is already available for actual use. Which process mining technique(s) is (are) used to generate the outputs? Which input should an instructor give? Can instructors customize the instrument? Maybe a short subsection at the end of section 5 could tackle these points.

** Evaluation **

- Line 287: Wouldn’t it make more sense to discuss the evaluation stages first and only discuss how evaluation data is collected and outcomes are analysed afterwards?

- Line 329-331: Here, it should be added that the focus is on the evaluation of the proposed taxonomy.

- Line 335: Does the “totally agree” refer to their response to (1) and (2) or to all three statements? In the former case, what about the third instructor?

- Line 347-353: The results are a bit mixed here. It would be good if the authors could highlight how this feedback could be incorporated here or in the conclusion.

** Minor points **

When proofreading the revised version of the manuscript, be aware of minor language inaccuracies. Some examples include:

- Line 87: “… is a taxonomy for [a/the] procedural skills training course.”  (or courses in plural)

- Line 93: “We generated a taxonomy [using?] the semantics [of?] the BPMN model…”

- Line 156: “… identifying and controlling [] an action step [correctly] each time …”

- Line 222: “Answer show identification …” > Not clear

- Line 283: “… to answer quantitative [should this be qualitative?] questionnaires writing in a paper…” > Just replace this by “written questionnaires” or even just questionnaires as I think they are by definition written?

- Line 315: “… she prefers global scales instead [of a] checklist because…”

Author Response

This paper focuses on the use of process mining to support medical training. It presents both a taxonomy of activities and a process-oriented instrument to provide feedback to students. The artefacts are evaluated by instructors who teach one particular procedure: the insertion of a central venous catheter.

After a thorough assessment of the paper, I can conclude that this paper constitutes a valuable contribution to the special issue. It covers a topic which translates process mining techniques to instruments which can be used in practice (in this case educational practice). There is a need for research in that direction. Even though the evaluation is based on the input of three instructors, I feel that this provides sufficient evidence that the proposed artefacts are valuable and, hence, should be developed further in the future. The only thing I miss in the current paper is a short subsection outlining which process mining techniques are used and how/to which extent the proposed instrument is actually available for use. Some other comments are listed below on a section-by-section basis, but these are not of a fundamental nature. I am confident that the authors can fix them within a reasonable timespan.

  • We appreciate the reviewer's comments and his/her assessment of our approach to address a Medical Education problem with Process Mining tools. We included information about the Process Mining techniques used in lines 153 - 156:

“Regarding Process Mining techniques, we used the ideas of algorithms based on Directly-Follows Graphs [18], which are commonly implemented in commercial tools (such as Disco and Celonis). We used this approach because it can be easily understood by non-experts users of Process Mining. Although this approach has some limitations such as a representative bias, it shows the behaviour of a process in a simple way [18].”

DETAILED REMARKS ON A SECTION-BY-SECTION BASIS

** Introduction **

- The authors might want to make it more explicit how process mining extends tools such as checklists or GRS in the fourth paragraph of the introduction. Adding an example of such an existing tool later in the paper could also help to understand the added-value of the proposed approach.

  • We added an example to clarify how Process Mining analysis offers tools such as checklists and GRS. We included this in lines 42 - 46:

“The information captured by an analysis of the sequence of steps through Process Mining adds information that is not possible to capture with a checklist, which is limited to a dichotomous assessment of the execution of specific steps of a procedure, without considering the order nor the unnecessary repetition. On the other hand, it explicitly explains a qualitative dimension included in the GRS, such as fluidity in the execution of the steps of a procedure.”

** Running case **

- Line 75-77: Readability would be enhanced if the sentence regarding event logs would be part of the same paragraph as the explanation how these event logs were created.

  • We moved the sentence regarding event logs to the paragraph that explains how the event logs were created. We included the modification with the sentences correctly ordered in lines 86 - 89:

“This software allowed us to generate event logs from videos recorded while students performed the procedure. Each event of the event log used contains a student as case identifier, a procedure step as activity and date time when the student performed the procedure step as the timestamp.”

- Line 81-82: How does the web-based software work? Is it based on manual labelling?

  • We uploaded the videos onto the web-based software (called POMElog). An expert tags each video on the platform using the tags displayed by the software. Labels are the activities defined in the process model of the procedure. Once the expert finishes tagging all the videos, the event log is exported. We included this in lines 86 - 90:

“We obtained data using a web-based software called POMElog [17]. This software allowed us to generate event logs from videos recorded while students performed the procedure. Each event of the event log used contains a student as case identifier, a procedure step as activity and date time when the student performed the procedure step as the timestamp. An expert manually tagged each video with the activities shown in the process model (see Figure 1). Once all videos were tagged, we obtained the event log used in this article.”

** Taxonomy of activities **

- Line 87-92: The goal and benefits of the taxonomy of activities should be clarified.

  • The taxonomy, like the instrument, helps to build and improve the mental model of the instructors. Both tools help instructors share information in a structured way and standardize the correct sequence of steps, helping to prevent instructors from missing information when teaching. We included this in lines 414 - 420:

“The taxonomy and the instrument can help instructors in building or enhancing their mental model. For instructors, it is challenging to share their mental model [26, 27], and that produce they omit close to 70% of the information needed by students during their learning process [25]. The taxonomy and the instrument help to address these problems: with the taxonomy, it is possible to share information about the steps of the procedure and the correct sequence of them; with the instrument, the information utilized to teach the correct sequence is standardized, avoiding the omission of information. Also, both tools help to give a structure as to how to teach the procedure, providing standardization of the contents and an objective way for assessment and giving feedback.”

- Line 97-117: An implicit order between activities is highlighted: preparation > identification > action > control. Wouldn’t it make sense to order them in the numbered list this way?

  • We picked up this suggestion also made by reviewer 1. We reordered points listed in lines 106 - 125 using the implicit order proposed by the taxonomy, and rewrote them less wordily to avoid reader confusion.

- This section would benefit from a bit of background information regarding how the flowchart in Figure 1 was created and validated.

  • We appreciate the reviewer's suggestion. We included this information in lines 79 - 82:

“To generate the process model, an initial model was generated using activities included in validated checklists. To avoid a biased model, [15] developed a Delphi panel with experts on the procedure through an online survey. They included the activities in the model when the panel reached 80% consensus for each activity, thus obtaining the ideal execution of the procedure.”

** Questions to discover undesired patterns **

- Line 144-145: Another undesired pattern would involve executing action activities in a sequential ordering relation in the wrong order. This is currently missing.

  • The execution of the action activities in a sequential wrong order is captured in the first section of the instrument (A1. Sequence of executed action activities). However, in the case of central venous catheter placement, these activities must be done in the proper order due to them being dependent on each other to some degree: for example, it is mandatory to “puncture” with the trocar for “Guidewire Install”, since it is impossible to pass the guide before “puncturing” with the trocar; after having the guidewire installed, you can "Advance Catheter" through it. For this reason, in this case, incorrect sequences will not be displayed in the instrument, which could be captured for other procedures that allow more freedom in execution.

- Line 152: It might be good to highlight why the excessive execution of identification/control steps is undesired.

  • We appreciate the reviewer's suggestion. We included this information in lines 186 - 187:

“An undesired pattern is the excessive execution of identification and control steps, because this indicates lack of fluidity and economy of movement [18].”

- The authors could consider to integrate sections 4 and 5, in which the current section 4 would become a subsection. Currently, it is not entirely clear what the purpose of section 4 as an autonomous section is.

  • We appreciate the reviewer's suggestion. We integrated both sections of the article creating one called “Discovering undesired patterns”. Furthermore, to improve the clarity and readability of the article, we changed the presentation of the information to show each question with its respective answer immediately. The new order of information is presented in lines 157 - 278. Additionally, in lines 143 - 156 we added introductory paragraphs to the section “Discovering undesired patterns”:

“In this section, we present the design of an instrument (see Figure 2) to discover undesired patterns. This instrument helps to do a retrospective analysis of the course performance regarding sequence errors. In addition, the instrument provides information at the course level and at the specific student level, allowing the comparison between both levels, knowing the overall performance of the course and thus planning the next sessions, either with current or future students.

The instrument contains answers for questions designed using the taxonomy presented in section 3. We remark that these questions did not allow us to analyse the successfulness of each isolated step, but helped us to discover undesired patterns related to the sequence of steps. Below, we present the four questions, each with its answer. We also generated the answers using the taxonomy, and then we put them together in the instrument shown in Figure 2.

Regarding Process Mining techniques, we used the ideas of algorithms based on Directly-Follows Graphs [18], which are commonly implemented in commercial tools (such as Disco and Celonis). We used this approach because of the ease of understanding they have by non-expert users of Process Mining. Although this approach has some limitations such as a representative bias, it shows the behaviour of a process in a simple way [18].”

** Discovering undesired patterns **

- Line 171: Maybe the authors should elaborate a bit more about what the instructor can learn at the two levels.

  • We appreciate the reviewer's suggestion. We included this information in lines 144 - 147:

“This instrument helps to do a retrospective analysis of the course performance regarding sequence errors. In addition, the instrument provides information at the course level and at the specific student level, allowing the comparison between both levels, knowing the overall performance of the course and thus planning the next sessions, either with current or future students.”

- Line 206-207: At the course level, the measure indicates the number of students who performed at least one identification/control activity. Is this entirely consistent with Q2, where it is highlighted that an undesired pattern is the excessive execution of identification/control activities?

  • We appreciate the reviewer's comment, who focuses on information that may be useful to the instructor, not captured in the current format of the instrument. We have included a paragraph listing possible improvements to the instrument that includes this suggestion in lines 429 - 432.

Future work to enhance interpretability of the instrument are improvements regarding repetitions or reworks of action activities (A1 of Figure 2), the excessive amount of identifications and control activities performed by the course in (A2 of Figure 2), and correctness of identification and control each time an action activity is performed (A3 of Figure 2).

- It would be interesting to know to what extent the proposed instrument is already available for actual use. Which process mining technique(s) is (are) used to generate the outputs? Which input should an instructor give? Can instructors customize the instrument? Maybe a short subsection at the end of section 5 could tackle these points.

As regards the Process Mining techniques we use, we included information about this in lines 153 - 156:

Regarding Process Mining techniques, we used the ideas of algorithms based on Directly-Follows Graphs [18], which are commonly implemented in commercial tools (such as Disco and Celonis). We used this approach because it can be easily understood by non-expert users of Process Mining. Although this approach has some limitations such as a representative bias, it shows the behaviour of a process in a simple way [18].

As regards the other questions, this research provides evidence that the instrument is capable of satisfying in an understandable and usable way the information needs of an instructor. At the moment, the instrument cannot be customized, and this possibility is subject to having the necessary input data, in this case the Event Log with the executions. We must emphasize that this approach is among the first publications that apply Process Mining tools to medical education problems. Therefore, as we can count on structured data obtained from the computer support system or through methods such as the one exposed in our research, it will be possible to make it available for use in other centers. The possibility of transferring this approach to Medical Education is being developed with research studies like the one we present.

** Evaluation **

- Line 287: Wouldn’t it make more sense to discuss the evaluation stages first and only discuss how evaluation data is collected and outcomes are analysed afterwards?

  • We agree with the reviewer in the suggested order. We reordered the paragraphs, first discussing the evaluation stages and then the outcomes and data collection. We included this change in lines 281 - 294.

“We conducted the evaluation through the following stages shown in Table 1. The focus of the first stage was to know the current ways instructors use to teach the sequence of steps, give feedback about the sequence, and assess whether students learnt the sequence. Then, we evaluated the ease of understanding and usefulness of taxonomy for typical instructor tasks, and the interpretability and usability of the instrument as well as the opinion of instructors after using the instrument with the instructors.

Table 1. Evaluation stages.

Stage

Task

1

Current teaching of the sequence of steps.

2

Taxonomy explanation to instructors and its usefulness.

3

Instrument explanation to instructors.

4

Instrument interpretability by instructors.

5

Usability analysis of the instrument.

6

Instructors’ opinion after using the instrument.

For evaluating the taxonomy and the instrument, we asked three experts who commonly teach the CVC installation. They teach in two institutions in anesthesiology and internal medicine specialities, have 5.7 years of experience as a CVC instructor on average, and 12.3 years of experience performing the procedure on average. Evaluation results were analyzed using qualitative content analysis [20]. We asked instructors to answer questionnaires writing in a paper, and open-ended questions were audio recorded. Then, we transcribed the audios and analysed them, to then create categories coding quotes to interpret the answers. We included participant quotes in the paper, and we identified them anonymously with I1, I2 and I3.”

- Line 329-331: Here, it should be added that the focus is on the evaluation of the proposed taxonomy.

  • We modified both lines mentioning that the focus is on evaluating the taxonomy. We included the change in lines 330 - 331:

“We evaluated the taxonomy by asking instructors about dimensions shown on Table 3 using a 5-point visual analogue scale, where one means ‘totally disagree’ and five ‘totally agree’.”

- Line 335: Does the “totally agree” refer to their response to (1) and (2) or to all three statements? In the former case, what about the third instructor?

  • We rewrote these lines to improve their comprehensibility. "Totally agree" refers to the response of 2 instructors to sentence 3 of table 3. We included the change in lines 333 - 334:

“Results show that all the instructors agreed with sentences nº1 and nº2 (see Table 3), while one instructor agreed and two totally agreed with sentence nº3 (see Table 3)”

- Line 347-353: The results are a bit mixed here. It would be good if the authors could highlight how this feedback could be incorporated here or in the conclusion.

  • We appreciate the reviewer's comment. We rewrote the results by first showing the questions in which the instructors performed well, and then showing the questions in which some instructors answered poorly. We included this change in lines 346 - 352:

“Results show that all instructors answered the majority of questions on Table 4 correctly (88.1% correct answers on average by each instructor, SD = 3.37%), confirming the success of this Interactive Pattern Recognition case [22]. All instructors answered all the questions related to the number of identification/control activities executed and procedure preparation correctly (answers A2 and A4, respectively). However, instructors answered some questions regarding repetitions or reworks of action activities incorrectly (two instructors answered question 1.1 incorrectly, one instructor answered question 1.3 wrongly), as well as questions related to the correct execution of identification and control (one instructor answered question 3.2 incorrectly, another instructor answered question 3.3 wrongly).”

  • Additionally, we added improvements to the instrument as future work in the discussion. The improvements are based on the results obtained in the interpretability test. We included this in lines 429 - 432, which is a paragraph included in response to another comment by the same reviewer.

** Minor points **

When proofreading the revised version of the manuscript, be aware of minor language inaccuracies. Some examples include:

  • The document was extensively revised by a native speaker translator, so grammar, spelling was checked, and verb tenses were unified into the past tense.

- Line 87: “… is a taxonomy for [a/the] procedural skills training course.”  (or courses in plural)

  • We included this change in line 95:

“The first contribution of this paper is a taxonomy for a procedural skills training.”

- Line 93: “We generated a taxonomy [using?] the semantics [of?] the BPMN model…”

  • We included this change in line 101:

“We generated a taxonomy using the semantic of the BPMN model provided by [14].”

- Line 156: “… identifying and controlling [] an action step [correctly] each time …”

  • We included this change in line 212:

“... identifying and controlling an action step correctly each time it is executed.”

- Line 222: “Answer show identification …” > Not clear

  • We included this change in line 222:

“A3 in Figure 2 ...”

- Line 283: “… to answer quantitative [should this be qualitative?] questionnaires writing in a paper…” > Just replace this by “written questionnaires” or even just questionnaires as I think they are by definition written?

  • This line was modified as previously answered to reviewer 1. We included the change in lines 291 - 292 of the new version of the article, writing:

“We asked instructors to answer questionnaires writing in a paper, and open-ended questions were audio recorded.”

- Line 315: “… she prefers global scales instead [of a] checklist because…”

  • We included this change in line 316:

“... she prefers global scales instead of a checklist because ...”

Reviewer 4 Report

The current paper deals with teaching of surgical procedures in medical training, using the ultrasound-guided insertion of a central venous catheter as a model. The paper is principally very interesting as it touches a field very important for future developments – teaching using the possibility of describing standards as the first step before using any tool for example also in the OR. Completely other way than used many years, when a student or an assistant had to take part many times before he got the possibility to interfere first time for himself at the patient. Congratulations.

Principally the paper is written in a scientific style. The figures and the tables are understandable and correctly. Especially the figures very good explain, what’s to do and when.

As the text is very long and expanding, I decided to interrupt my reading and send the paper back to the editors. From my point of view is must be possible to explain the aim and the content of this work much more dense and clearly. If the authors could shorten the paper, I would be willing to review it again. In the present form I regret - but finally I´ve gave up. I´m sorry, but in the current form I cannot recommend the paper for publication.

Author Response

The current paper deals with teaching of surgical procedures in medical training, using the ultrasound-guided insertion of a central venous catheter as a model. The paper is principally very interesting as it touches a field very important for future developments – teaching using the possibility of describing standards as the first step before using any tool for example also in the OR. Completely other way than used many years, when a student or an assistant had to take part many times before he got the possibility to interfere first time for himself at the patient. Congratulations.

Principally the paper is written in a scientific style. The figures and the tables are understandable and correctly. Especially the figures very good explain, what’s to do and when.

  • We appreciate the reviewer's comments. Our intention is to deliver tools in pre-clinical stages that help consolidate technical skills that ensure safe and quality care for patients.

As the text is very long and expanding, I decided to interrupt my reading and send the paper back to the editors. From my point of view is must be possible to explain the aim and the content of this work much more dense and clearly. If the authors could shorten the paper, I would be willing to review it again. In the present form I regret - but finally I´ve gave up. I´m sorry, but in the current form I cannot recommend the paper for publication.

  • We regret that the development of this article is longer or more extensive than the reviewer considers appropriate. However, we believe it is necessary to take into account that we followed the structure required by this special issue, which requires extending the initial article presented in the “Process Oriented Data Science for Healthcare 2019 (PODS4H)” workshop by 30%. Furthermore, the specific theme of this special issue is Process Mining applied to health. Given that it is an interdisciplinary investigation, we thought it important to provide enough background to allow both researchers in the Medical Education area and the Process Mining area to better understand our approach and the results obtained. We hope that the changes made to respond to the reviewers' suggestions increase the clarity and readability of our article, so that it can be read by researchers from both disciplines.

Round 2

Reviewer 1 Report

Line 36- Something missing in this sentence.

Lines 37-39- There are other writing errors here too.

Line 46- replace  On the other hand with Conversely

Lines 52-54- A quantitative or qualitative evaluation doesn't ask anything. You use an instrument that asks questions. You are using mixed-methods including qualitative and quantitative methods to analyze the data collected from the instrument.

Line 110- How many sites were possible. Was the study restricted to internal jugular cannulation or were some subclavian lines or femoral lines? If multiple sites were used in the study, the results may justify the procedures for all sites. If not, that is ok but it may only validate one central line site.

Lines 201-202- The sentence added needs to be rewritten.

Line 333-334- One instructor agreed and two totally agreed. Could this be confusing? You may want to switch the two around and write, two strongly agreed, while one agreed.

Line 443- Consider adding- Further studies should be done after instructors gain experience using the tool. 

Check for the proper positioning of footnotes.

Author Response

The authors are grateful to the anonymous reviewers for their constructive and insightful comments.Following suggestions, we have answered to each of the reviewers' comments, point by point, pointing out the lines in which the change was made in the article and the exact change.

Line 36- Something missing in this sentence.

  • This line was modified. We changed the words to clarify the use of Process Mining in the study cited. This change is in lines 35 – 37:

“Lira et al [9] showed the use of Process Mining to give specific feedback, using data obtained from recorded executions of a procedure performed by students.”

Lines 37-39- There are other writing errors here too.

  • These lines were modified, correcting writing errors. This change is in lines 37 – 39:

“Also, de la Fuente et al [10] used Process Mining to compare trainees and experts with the ideal sequence of steps. However, both studies did not emphasize information relevant to instructors and are difficult for them to interpret.”

Line 46- replace  On the other hand with Conversely

  • This line was modified, replacing in line 45 “On the other hand” with “Conversely”.

Lines 52-54- A quantitative or qualitative evaluation doesn't ask anything. You use an instrument that asks questions. You are using mixed-methods including qualitative and quantitative methods to analyze the data collected from the instrument.

  • We agree with the reviewer on the methodology we used to analyze the data (mixed-methods), and we indeed used instruments to collect data. To make this clearer in the article, we added corrections to the writing. The change is in lines 53 - 55:

“(2) an evaluation of the taxonomy’s usefulness through open-ended questions, (3) an evaluation of the instrument’s interpretability by means of a questionnaire, (4) an evaluation of the instrument's usability using the System Usability Scale (SUS) [24] and (5) the opinion of instructors about the instrument after using it.”

Line 110- How many sites were possible. Was the study restricted to internal jugular cannulation or were some subclavian lines or femoral lines? If multiple sites were used in the study, the results may justify the procedures for all sites. If not, that is ok but it may only validate one central line site.

  • Our approach was carried out in an ultrasound-guided internal jugular central venous access training course, so the model's activities refer to this procedure. In the description of the procedure we were not clear enough, so the references to the procedure were modified by replacing “Central Venous Catheter (CVC) procedure” with “Ultrasound-Guided Internal Jugular Central Venous Catheter (UGIJCVC) placement”, in section 2.1. CVC case and on the rest of the article.

Lines 201-202- The sentence added needs to be rewritten.

  • We modified this line. We made some writing corrections, and the change is in lines 201 – 202:

“after ‘Advance catheter’ and ‘Remove guidewire’, but the best approach is to do two control activities”.

Line 333-334- One instructor agreed and two totally agreed. Could this be confusing? You may want to switch the two around and write, two strongly agreed, while one agreed.

  • We modified this line. This change is in lines 333 – 334, writing “Also, two instructors totally agreed and one agreed with sentence nº3”.

Line 443- Consider adding- Further studies should be done after instructors gain experience using the tool. 

  • We appreciate the reviewer’s comment. We added the suggestion in line 445 writing “Further studies should be done once instructors gain experience using the tool”.

Check for the proper positioning of footnotes.

  • We appreciate the reviewer’s comment. In the final version of the article, we removed the footnotes, including the corresponding notes in the content of the article.

Reviewer 2 Report

Authors have included satisfactory responses. All comments have been replies in a right way and changes have been introduced improving the manuscript.

Author Response

Authors have included satisfactory responses. All comments have been replies in a right way and changes have been introduced improving the manuscript.

  • We appreciate the reviewer’s comments, and we are happy to have provided satisfactory responses to their comments and suggestions.